# Comparative Benefits and Risks Associated with Currently Authorized COVID-19 Vaccines

**DOI:** 10.3390/vaccines10122065

**Published:** 2022-12-01

**Authors:** Jacob R. Albers, Jeffrey B. Brown, Shaun V. Charkowick, Natasha Ram, Farina A. Klocksieben, Ambuj Kumar

**Affiliations:** 1Morsani College of Medicine, University of South Florida, Tampa, FL 33620, USA; 2Research Methodology and Biostatistics Core, Office of Research, Department of Internal Medicine, 12901 Bruce B. Downs Blvd MDC 27, Tampa, FL 33612, USA

**Keywords:** COVID, vaccines, grade, evidence, transparency

## Abstract

This article provides a systematic assessment of the efficacy, risks, and methodological quality of evidence from five major publicly available vaccine trials. Results from Pfizer-BioNTech mRNA, Moderna-US NIH mRN-1273, AstraZeneca-Oxford ChAdOx1 nCov-19, Gamaleya GamCovidVac (Sputnik V), and Ad26.COV2.S Johnson & Johnson vaccines were included. Extracted benefits and risks data from each trial were summarized using the GRADE approach denoting the overall certainty of evidence along with relative and absolute effects. Relative risk reduction across all five vaccine trials ranged from 45% to 96%. Absolute risk reduction in symptomatic COVID-19 ranged from 6 to 17 per 1000 across trials. None of the vaccines were associated with a significant increase in serious adverse events compared to placebo. The overall certainty of evidence varied from low to moderate. All five vaccines are effective and safe, but suggest room for improvement in the conduct of large-scale vaccine trials. Certainty of evidence was downrated due to risk of bias, which can be mitigated by improving transparency and thoroughness in conduct and reporting of outcomes.

## 1. Introduction

Several public health strategies, such as mask mandates, isolation, social distancing, and contact tracing, aimed at reducing transmission have been insufficient to stop the global rise of COVID-19 [1]. A widely available vaccine has been suggested as the most effective method to prevent the rapid spread of infections. At present, 50 COVID-19 vaccines have reached the final stages of clinical trial testing in humans [2]. First to emerge publicly, the clinical trial results from five major vaccines have been published outlining their efficacy and safety: Pfizer-BioNTech BNT162b2 mRNA vaccine [3], Moderna-US National Institute of Health (NIH) mRN-1273 vaccine [4], Johnson & Johnson (J&J) Ad26.COV2.S vaccines [5], AstraZeneca-Oxford ChAdOx1 nCov-19 vaccine [6], and Gamaleya GamCovidVac (Sputnik V) vaccine [7]; Pfizer, Moderna, J&J, AstraZeneca, Sputnik, respectively. Of these, Pfizer was the first to be granted full FDA approval in the United States [8]. Despite wide dissemination of the benefits of these vaccines, 76% of the United States and less than 63.5% of the global population are currently vaccinated [9,10,11,12].

Multiple factors have attributed to low vaccination rates globally. In the United States, vaccine hesitancy remains a major issue despite widespread availability and free access. One contributing factor may be the lack of comparative information related to benefits and risks associated with various vaccines. Specifically, reliable evidence related to the efficacy of all COVID-19 vaccines is critical for all stakeholders, including patients, physicians, healthcare workers, and policymakers. Furthermore, accurate interpretation of this evidence is required to make effective personal and public health decisions regarding a COVID-19 vaccine. Unfortunately, misinterpretation and misrepresentation of clinical trial data have been a common occurrence [13,14]. It is imperative that the benefits and risks associated with available vaccines be communicated transparently and concisely to allow for informed choices. The goal of this paper is to provide a systematic assessment of the risks, efficacy, and methodological quality of evidence from five major vaccine trials in a format that may be used as a tool to facilitate informed decision making for all stakeholders involved in the COVID-19 vaccination efforts.

## 2. Materials and Methods

We compiled the interim results of five COVID-19 vaccine studies published in scientific journals and that are currently globally distributed: Pfizer-BioNTech BNT162b2 mRNA vaccine [3], Moderna-US National Institute of Health (NIH) mRN-1273 vaccine [4], AstraZeneca-Oxford ChAdOx1 nCov-19 vaccine [6], Gamaleya GamCovidVac (Sputnik V) vaccine [7], and Johnson & Johnson (J&J) Ad26.COV2.S vaccines [5]. In addition to the published reports, FDA briefing documents were referenced for the Pfizer, Moderna, and J&J vaccines. Metrics of efficacy and adverse events were selected based on saliency and homogeneity of criteria definition across trials. Data on benefits and risks associated with each vaccine were extracted from these reports. The efficacy endpoints were incidence of symptomatic COVID-19 cases, severe COVID-19 cases, mortality due to COVID-19, and all-cause mortality. Adverse events included any serious adverse events due to vaccination and any related unsolicited adverse events due to vaccination defined according to each study’s criteria.

The abstracted benefits and risks data from each trial were summarized using the GRADE approach denoting the relative and absolute effects along with the overall methodological quality of evidence associated with each outcome [15]. All analyses for the desirable outcomes (i.e., benefits) were performed using the intention to treat principle (ITT). All adverse events were analyzed using a per-protocol (PP) approach.

## 3. Results

The trial and participant characteristics for five vaccines are reported in the Appendix A. Briefly, all five studies were randomized controlled trials (RCT) and enrolled between 15,210 to 44,325 subjects. Further summary of trial characteristics is provided in Appendix B. Vaccine, Trial, and Participant Characteristics

Outcomes:

Sputnik (ITT population = 21,977 participants)

Assessment of COVID-19 Positivity

Symptomatic COVID-19 was defined as a positive PCR test in addition to clinical signs of respiratory infection. Seventy-nine (0.5%) participants in the vaccine arm and ninety-six (1.8%) in the placebo arm reported symptomatic COVID-19 (relative risk [RR] 0.27; 95% confidence intervals [CI] 0.20–0.37; Appendix A). The overall certainty of evidence was moderate (Appendix A).

Symptomatic COVID-19 After Effective Period

The effective period for the Sputnik vaccine was 21 days after the first immunization. Sixteen cases (0.1%) in the vaccine arm and sixty-two (1.1%) in the placebo arm were reported after the effective period (RR 0.09; 95% CI 0.05–0.15; Appendix A). The overall certainty of evidence was moderate (Appendix A).

Severe COVID-19 Cases After Effective Period

Severe COVID-19 cases were reported combined with moderate COVID-19 cases for all participants who received both injections. Zero (0.0%) participants in the vaccine arm contracted moderate-severe COVID-19 compared to twenty (0.4%) in the placebo arm (RR 0.01; 95% CI 0.00–0.13; Appendix A). The overall certainty of the evidence was moderate (Appendix A).

COVID-19-Related Mortality

Two (0.01%) deaths in the vaccine arm and zero (0.0%) deaths in the placebo arm were reported (RR 1.16; 95% CI 0.08–34.56; Appendix A). Both participants who died displayed symptoms within 5 days of their first vaccination and had significant comorbidities. The overall certainty of the evidence was low (Appendix A).

All-Cause Mortality

Three (0.02%) participants in the vaccine arm and one (0.02%) participant in the placebo arm died throughout the trial (RR 1.00; 95% CI 0.10–9.57; Appendix A). The overall certainty of the evidence was low (Appendix A).

Any Serious Adverse Events

Serious adverse events were any reaction that required hospital admission. Forty-three (0.3%) of participants in the vaccine arm and twelve (0.2%) in the placebo arm reported serious adverse events (RR 1.19; 95% CI 0.63–2.25; Appendix A). The overall certainty of the evidence was low (Appendix A).

Related Unsolicited Serious Adverse Events

An independent data monitoring committee (IDMC) determined whether serious adverse events were related to vaccination. The IDMC determined that zero (0.0%) participants in the vaccine arm and one (0.02%) in the placebo arm experienced an unsolicited serious adverse event related to the vaccination (RR 0.11; 95% CI 0.00–2.72; Appendix A). The overall certainty of the evidence was low (Appendix A).

AstraZeneca (ITT population = 23,848 participants)

Assessment of COVID-19 Positivity

Symptomatic COVID-19 was defined as a positive PCR test in addition to one of the following: Fever ≥ 37.8 °C, cough, shortness of breath, anosmia or ageusia. Sixty-three (0.5%) participants in the vaccine arm and one hundred and fifty (1.3%) participants in the placebo arm developed symptomatic COVID-19 (RR 0.41; 95% CI 0.31–0.55; Appendix A). The overall certainty of the evidence was low (Appendix A).

Symptomatic COVID-19 after Effective Period

The effective period for the AstraZeneca vaccine was 14 days after the second immunization. Thirtyseven (0.3%) cases in the vaccine arm and one hundred and twelve (1.0%) in the placebo arm were reported (RR 0.32; 95% CI 0.22–0.47; Appendix A). The overall certainty of the evidence was low (Appendix A).

Severe COVID-19 Cases after Effective Period

Severe COVID-19 was defined as WHO clinical progression score of ≥6. Zero (0.0%) participants in the vaccine arm contracted severe COVID-19 after the effective period compared to one (0.01%) in the placebo arm (RR 0.32; 95% CI 0.01–7.97; Appendix A). The overall certainty of the evidence was very low (Appendix A).

COVID-19-Related Mortality

COVID-19-related mortality was zero (0.0%) in the vaccine arm and one (0.01%) in the placebo arm (RR 0.33; 95% CI 0.01–7.98; Appendix A). The overall certainty of the evidence was low (Appendix A).

All-Cause Mortality

One (0.01%) participant in the vaccine arm and four (0.03%) in the placebo arm died (RR 0.24; 95% CI 0.03–2.18; Appendix A). The overall certainty of the evidence was low (Appendix A).

Any Serious Adverse Events

The definition of serious adverse events was unclear. Seventy-seven (0.6%) participants in the vaccine arm and seventy-eight (0.7%) in the placebo arm reported any serious adverse events (RR of 0.96; 95% CI 0.70–1.32; Appendix A). The overall certainty of the evidence was low (Appendix A).

Related Unsolicited Serious Adverse Events

Causality of adverse events was determined by a site investigator. One (0.01%) participant in the vaccine arm and one (0.01%) in the placebo arm experienced an unsolicited serious adverse event related to the vaccination (RR 0.98; 95% CI 0.06–15.59; Appendix A). The overall certainty of the evidence was low (Appendix A).

Moderna (ITT population = 30.420 participants)

Assessment of COVID-19 Positivity

Symptomatic COVID-19 was defined as a positive PCR test in addition to two of the following: Fever, chills, myalgia, headache, sore throat, loss of taste or smell or one of the following: Respiratory sign or symptom. Nineteen (0.1%) participants in the vaccine arm and two hundred and sixty-nine (1.8%) in the vaccine arm had symptomatic COVID-19 (RR 0.07; 95% CI 0.04–0.11; Appendix A). The overall certainty of the evidence was moderate (Table 1).

Symptomatic COVID-19 After Effective Period

The effective period for the Moderna vaccine was 14 days after the second immunization. Twelve (0.1%) cases in the vaccine arm and two hundred and four (1.3%) in the placebo arm were reported (RR 0.06; 95% CI 0.03–0.11; Appendix A). The overall certainty of the evidence was moderate (Table 1).

Severe COVID-19 Cases After Effective Period

Severe COVID-19 was defined as symptomatic COVID-19 in addition to death, ICU admission or severe respiratory, cardiac, renal, hepatic or neurologic symptoms. Zero (0.0%) participants in the vaccine arm contracted severe COVID-19 after the effective period compared to thirty (0.2%) in the placebo arm (RR 0.02; 95% CI 0.00–0.27; Appendix A). The overall certainty of the evidence was moderate (Table 1).

COVID-19-Related Mortality

Zero (0.0%) deaths in the vaccine arm and one (0.01%) in the placebo arm were reported (RR 0.33; 95% CI 0.01–8.18; Appendix A). The overall certainty of the evidence was moderate (Table 1).

All-Cause Mortality

Two (0.01%) participants in the vaccine arm and four (0.03%) in the placebo arm died throughout the trial (RR 0.50; 95% CI 0.09–2.73; Appendix A). The overall certainty of the evidence was moderate (Table 1).

Any Serious Adverse Events

Serious adverse events were defined as any adverse event that was life-threatening, a medically important event or caused significant incapacity, hospitalization or death. One hundred and forty-seven (1.0%) participants in the vaccine arm and one hundred and fifty-three (1.0%) in the placebo arm reported serious adverse events during the trial (RR 0.96; 95% CI 0.77–1.20; Appendix A). The overall certainty of the evidence was moderate (Table 1).

Related Unsolicited Serious Adverse Events

Unsolicited serious adverse events were any side effects not specifically inquired about. Six (4.0%) participants in the vaccine arm and four (0.03%) in the placebo arm experienced unsolicited adverse events related to the vaccination (RR 1.50; 95% CI 0.42–5.31; Appendix A). The overall certainty of the evidence was moderate (Table 1).

Pfizer (ITT population = 43,448 participants)

Assessment of COVID-19 Positivity

Symptomatic COVID-19 was defined as a positive PCR test in addition to one of the following: Fever, new or increased cough, new or increased shortness of breath, chills, new or increased muscle pain, new loss of taste or smell, sore throat, diarrhea or vomiting. Fifty (0.2%) participants in the vaccine arm and two hundred and seventy-five (1.3%) in the vaccine arm had symptomatic COVID-19 (RR 0.18; 95% CI 0.13–0.25; Appendix A). The overall certainty of the evidence was moderate (Table 2).

Symptomatic COVID-19 after Effective Period

The effective period for the Pfizer vaccine was 7 days after the second immunization. Eight (0.04%) cases in the vaccine arm and one hundred and sixty-two (0.7%) in the placebo were reported (RR 0.05; 95% CI 0.02–0.10; Appendix A). The overall certainty of the evidence was moderate (Table 2).

Severe COVID-19 Cases after Effective Period

Severe COVID-19 was defined as symptomatic COVID-19 in addition to death, ICU admission, shock, severe systemic illness or severe respiratory, neurologic, hepatic or renal symptoms. One (0.005%) participant in the vaccine arm and three (0.01%) in the placebo arm contracted severe COVID-19 after the effective period (RR of 0.33; 95% CI 0.03–3.21; Appendix A). The overall certainty of the evidence was low (Table 2).

COVID-19-Related Mortality

Zero (0.0%) deaths in the vaccine and placebo arm were reported (Appendix A). The overall certainty of the evidence was high (Table 2).

All-Cause Mortality

Two (0.01%) participants in the vaccine arm and four (0.02%) in the placebo arm died throughout the trial (RR 0.50; 95% CI 0.09–2.73; Appendix A). The overall certainty of the evidence was moderate (Table 2).

Any Serious Adverse Events

Serious adverse events were defined as any adverse event that was life-threatening, resulted in prolonged disability, required hospital admission or death. One hundred and twenty-six (0.7%) participants in the vaccine arm and one hundred and eleven (0.6%) in the placebo arm reported serious adverse events during the trial (RR 1.13; 95% CI 0.88–1.46; Appendix A). The overall certainty of the evidence was moderate (Table 2).

Related Unsolicited Serious Adverse Events

Four (0.02%) unsolicited serious adverse events in the vaccine arm and zero (0.0%) in the placebo arm were reported (RR 8.99; 95% CI 0.48–167.03; Appendix A). The overall certainty of the evidence was moderate (Table 2).

Johnson & Johnson (ITT population = 44,325 participants)

Assessment of COVID-19 Positivity

Symptomatic COVID-19 was defined as a positive PCR test in addition to one of the following: Fever, sore throat, malaise, headache, myalgia, gastrointestinal symptoms, cough, chest congestion, runny nose, wheezing, skin rash, eye irritation or discharge, chills, new or changing olfactory or taste disorders, red or bruised looking feet or toes or shaking chills or rigors. One hundred and ninety-five (0.9%) participants in the vaccine arm and four hundred and thirty-five (2.0%) in the placebo had symptomatic COVID-19 (RR 0.45; 95% CI 0.38–0.53; Appendix A). The overall certainty of the evidence was moderate (Table 3).

Symptomatic COVID-19 after Effective Period

The effective period for the Johnson & Johnson vaccine was 28 days after immunization. The trial recorded sixty-six (0.3%) cases in the vaccine arm and one hundred and ninety-five (0.9%) in the placebo after the effective period (RR 0.34; 95% CI 0.26–0.45; Appendix A). The overall certainty of the evidence was moderate (Table 3).

Severe COVID-19 Cases after Effective Period

The trial reported severe-critical COVID-19 which we categorized as severe COVID-19 infection for comparison with the other trials. Severe-critical COVID-19 was defined as a positive PCR test in addition to one of the following: Death, ICU admission, shock or severe respiratory, cardiac, renal, hepatic or neurologic symptoms. Five (0.02%) participants in the vaccine arm and thirty-four (0.2%) in the placebo contracted severe-critical COVID-19 after the effective period (RR 0.15; 95% CI 0.06–0.38; Appendix A). The overall certainty of the evidence was low (Table 3).

COVID-19-Related Mortality

Zero (0.0%) deaths in the vaccine arm and five (0.02%) in the placebo arm were reported (RR 0.09; 95% CI 0.01–1.64; Appendix A). The overall certainty of the evidence was low (Table 3).

All-Cause Mortality

Three (0.01%) participants in the vaccine arm and sixteen (0.07%) in the placebo died throughout the trial (RR 0.19; 95% CI 0.05–0.64; Appendix A). The overall certainty of the evidence was low (Table 3).

Any Serious Adverse Events

Serious adverse events were defined as any adverse event that was life-threatening, transmitted by medical machinery, resulted in prolonged disability, required hospital admission, death or was determined to be medically important based on investigator judgement. Eighty-three (0.4%) participants in the vaccine arm and ninety-six (0.4%) in the placebo arm reported serious adverse events during the trial (RR 0.86; 95% CI 0.64–1.16; Appendix A). The overall certainty of the evidence was low (Table 3).

Related Unsolicited Serious Adverse Events

Seven (0.03%) participants in the vaccine arm and two (0.01%) in the placebo experienced unsolicited serious adverse events related to the vaccination (RR 3.50; 95% CI 0.73–16.84; Appendix A). The overall certainty of the evidence was low.

## 4. Discussion

The findings from RCTs assessing the efficacy of vaccines for the prevention of COVID-19 shows that all five vaccines were effective in preventing symptomatic COVID-19 infection. Relative risk reduction across all five vaccine trials ranged from 45% to 96%. Absolute risk reduction in symptomatic COVID-19 ranged from 6 to 17 per 1000 and was not associated with a significant increase in serious adverse events compared to placebo. The overall certainty of evidence was low to moderate across the included trials. This overall efficacy was similar for any time after the first dose and after the pre-determined effective periods for each vaccine (Appendix A). While J&J, Sputnik, and Moderna were associated with statistically significant reductions in risk of developing severe COVID-19 after the effective period, Pfizer and AstraZeneca were not, which is most likely due to the fact that the outcome is secondary, and thus not powered to detect this small difference (Appendix A). There were zero COVID-19 deaths in the vaccine arms of all trials except for Sputnik (Appendix A). All five vaccines were associated with non-significant reductions in COVID-19-related deaths and all-cause mortality with two exceptions; Sputnik vaccine was associated with a non-significant increase in COVID-19-related deaths in the vaccine group due to a failure of pre-trial COVID-19 screening, and Pfizer had zero COVID-19-related deaths (Appendix A). J&J was the only vaccine associated with a statistically significant reduction in all-cause mortality (Appendix A).

The benefit and safety profile favored all vaccines. However, the certainty of evidence varied from very low, low, moderate or high across vaccine trials (Table 1, Table 2 and Table 3, Appendix A). The certainty of evidence was downrated due to mostly imprecision and risk of bias for multiple outcomes across all vaccines. Main reasons for risk of bias included primary and secondary outcomes that were reported following the per-protocol approach (Sputnik, Moderna, Pfizer) and data that were not available for all participants vaccinated (J&J) or randomized (AstraZeneca, J&J). Primary reasons for imprecision were wide confidence intervals or those indicating the possibility of no effect, possibly due to a low number of events in the placebo arms. Furthermore, another important consideration is the heterogeneity in outcome definitions across vaccines. For example, the Sputnik trial defined severe COVID-19 as a positive PCR test in addition to severe systemic or respiratory symptoms. Meanwhile, the Moderna, Pfizer, and Johnson & Johnson trials all had definitions of severe COVID-19 that encompassed the Sputnik trial and included participants that had a positive PCR with severe neurologic, cardiac, hepatic or renal symptoms. Another example of trial heterogeneity is in the number of doses and length of time delineating the ‘effective period’ of each vaccine. Johnson & Johnson was the only single dose vaccine with a 28-day effective period. For two dose vaccines, the effective vaccine period varied across trials: Day 1 of second vaccination (Sputnik), day 7 (Pfizer), and day 14 (Moderna and AstraZeneca). To facilitate public health decision making, we suggest that future clinical trials engage in a more transparent reporting process by providing all outcomes after the first dose, after the second dose (if applicable), and after the designated effective period. For the interim, we encourage our readers to consider efficacy in the manner reported in this paper.

On a note of caution, the results do not provide answers on the comparative efficacy of vaccines as that would require conducting head-to-head clinical trials. Since each clinical trial was conducted within unique populations and time frames, unaccountable covariates may have exerted influence in unforeseen ways both in the magnitude and direction of effect. For this reason, drawing conclusions about the relative efficacy of each vaccine represents a transitive fallacy, which has been logically demonstrated as an important consideration for interpreting multiple clinical trials that investigate a given class of interventions [16]. We emphatically caution against comparing the relative efficacy of different vaccine trials with the current available evidence, which would require a different methodology and study design.

The understanding of COVID-19 and the vaccination effort is a rapidly evolving field with many clinical trials being performed and a multitude of scientific literature rapidly being published. This analysis is limited by the number of studies included into the GRADE evidence assessment. Further limitations to our analysis stem from the heterogeneity of study design and outcome definitions across each trial assessed. Specifically, the inclusion criteria for the multiple variables assessed among trials differed slightly between trials. Careful and detailed evaluation of the study protocols were required to effectively group comparable variables for our analysis. Another example of trial heterogeneity is in the number of doses and length of time delineating the ‘effective period’ of each vaccine. Johnson & Johnson was the only single dose vaccine with a 28-day effective period. For two dose vaccines, the effective vaccine period varied across trials: Day 1 of second vaccination (Sputnik), day 7 (Pfizer), and day 14 (Moderna and AstraZeneca). To facilitate public health decision making, we suggest that future clinical trials engage in a more transparent reporting process by providing all outcomes after the first dose, after the second dose (if applicable), and after the designated effective period for each trial.

Our analysis confirmed that all five vaccines were safe with no significant association between vaccination and adverse events across all trials. These results should facilitate shared decision making and possibly help in addressing vaccine hesitancy [17]. Additionally, adverse events that occurred are rare. The incidence of Guillain Barré syndrome (GBS) as an example is 5.8 per million doses of Pfizer, AstraZeneca or Moderna compared [18] to 479 per million COVID-19 positive cases [19]. Similarly, thrombotic events of 3.83 cases per million doses [20] of J&J versus 0.08% of COVID-19 positive patients equate to 80,000 per million [21]. Myocarditis was observed in 2.3 per 100,000 persons among people with one dose of mRNA vaccines versus estimates as high as 15% in COVID-19 positive patients [22], and additional widespread accounts of myocardial injury in hospitalized COVID-19 patients [23]. Since vaccination decreases the risk of COVID-19 infection, it also provides an absolute decrease in the risk of the above-mentioned pathologies that have been identified as associated with the vaccines.

In summary, the five vaccines evaluated in our paper are safe and effective for preventing COVID-19. Our analyses suggest room for improvement in the conduct and reporting the results of large-scale clinical trials to better facilitate shared decision making. When multiple clinical trials assessing the efficacy of a given intervention are anticipated, a uniform simplified standard of data reporting may be warranted. One possibility could include reporting outcomes utilizing both the intention-to-treat and per-protocol methods after each dose and after the designated effective period for each vaccine. In conjunction with clear public communication, this suggested that transparent reporting may reduce vaccine hesitancy and improve trust in health institutions [24]. Notwithstanding room for improvement, the evidence from these trials is sufficiently strong to suggest that, from a public health standpoint, there should be no bias in terms of which vaccines to utilize. Rather, vaccination initiatives should be chosen based on availability, accessibility, public acceptance, and cost.

## Figures and Tables

**Table 1 vaccines-10-02065-t001:** GRADE evidence profile denoting the benefits and risks associated with the Moderna vaccine for the prevention of COVID-19 in adults. The table includes relative and absolute effects along with the overall certainty of the evidence.

Certainty Assessment							Summary of Findings				
Participants (Studies) Follow-Up	Risk of Bias	Inconsistency	Indirectness	Imprecision	Publication Bias	Overall Certainty of Evidence	Study Event Rates (%)		Relative Effect (95% CI)	Anticipated Absolute Effects	
							With Placebo	With Moderna		Risk with Placebo	Risk Difference with Moderna
Any symptomatic cases											
30,420 (1 RCT)	Serious ^a^	not serious	not serious	not serious	none	⨁⨁⨁◯ ^d^ Moderate	269/15,210 (1.8%)	19/15,210 (0.1%)	RR 0.07 (0.04 to 0.11)	18 per 1000	16 fewer per 1000 (from 17 fewer to 16 fewer)
Symptomatic cases after effective period											
30,420 (1 RCT)	Serious ^a^	not serious	not serious	not serious	none	⨁⨁⨁◯ Moderate	204/15,210 (1.3%)	12/15,210 (0.1%)	RR 0.06 (0.03 to 0.11)	13 per 1000	13 fewer per 1000 (from 13 fewer to 12 fewer)
Severe cases after effective period											
30,420 (1 RCT)	Serious ^a^	not serious	not serious	not serious	none	⨁⨁⨁◯ Moderate	30/15,210 (0.2%)	0/15,210 (0.0%)	RR 0.02 (0.00 to 0.27)	2 per 1000	2 fewer per 1000 (from 1 fewer to --)
COVID-19 deaths											
30,420 (1 RCT)	not serious	not serious	not serious	Serious ^b,c^	none	⨁⨁⨁◯ Moderate	1/15,210 (0.0%)	0/15,210 (0.0%)	RR 0.33 (0.01 to 8.18)	0 per 1000	0 fewer per 1000 (from 0 fewer to 0 fewer)
All-cause mortality											
30,420 (1 RCT)	not serious	not serious	not serious	Serious ^b,c^	none	⨁⨁⨁◯ Moderate	4/15,210 (0.0%)	2/15,210 (0.0%)	RR 0.50 (0.09 to 2.73)	0 per 1000	0 fewer per 1000 (from 0 fewer to 0 fewer)
Any serious adverse events											
30,351 (1 RCT)	not serious	not serious	not serious	Serious ^b^	none	⨁⨁⨁◯ Moderate	153/15,170 (1.0%)	147/15,181 (1.0%)	RR 0.96 (0.77 to 1.20)	10 per 1000	0 fewer per 1000 (from 2 fewer to 2 more)
Related unsolicited serious adverse events											
30,351 (1 RCT)	not serious	not serious	not serious	Serious ^b^	none	⨁⨁⨁◯ Moderate	4/15,170 (0.0%)	6/15,181 (0.0%)	RR 1.50 (0.42 to 5.31)	0 per 1000	0 fewer per 1000 (from 0 fewer to 1 more)

CI: Confidence interval; RR: Risk ratio. Explanations: ^a^. The primary and secondary outcomes were reported following the per-protocol analysis approach. ^b^. The 95% confidence intervals for this outcome were wide. ^c^. The 95% confidence interval for this outcome includes the possibility of no effect. ^d.^ The overall certainty of evidence is visually delineated on a four-point scale.

**Table 2 vaccines-10-02065-t002:** GRADE evidence profile denoting the benefits and risks associated with Pfizer vaccine for the prevention of COVID-19 in adults. The table includes relative and absolute effects along with the overall certainty of the evidence.

Certainty Assessment							Summary of Findings				
Participants (Studies) Follow-Up	Risk of Bias	Inconsistency	Indirectness	Imprecision	Publication Bias	Overall Certainty of Evidence	Study Event Rates (%)		Relative Effect (95% CI)	Anticipated Absolute Effects	
							With Placebo	With Pfizer		Risk with Placebo	Risk Difference with Pfizer
Any symptomatic cases											
43,448 (1 RCT)	Serious ^a^	not serious	not serious	not serious	none	⨁⨁⨁◯ ^d^ Moderate	275/21,728 (1.3%)	50/21,720 (0.2%)	RR 0.18 (0.13 to 0.25)	13 per 1000	10 fewer per 1000 (from 11 fewer to 9 fewer)
Symptomatic cases after effective period											
43,448 (1 RCT)	Serious ^a^	not serious	not serious	not serious	none	⨁⨁⨁◯ Moderate	162/21,728 (0.7%)	8/21,720 (0.0%)	RR 0.05 (0.02 to 0.10)	7 per 1000	7 fewer per 1000 (from 7 fewer to 7 fewer)
Severe cases after effective period											
43,448 (1 RCT)	Serious ^a^	not serious	not serious	Serious ^b,c^	none	⨁⨁◯◯ Low	3/21,728 (0.0%)	1/21,720 (0.0%)	RR 0.33 (0.03 to 3.21)	0 per 1000	0 fewer per 1000 (from 0 fewer to 0 fewer)
COVID-19 deaths											
43,448 (1 RCT)	not serious	not serious	not serious	not serious	none	⨁⨁⨁⨁ High	0/21,728 (0.0%)	0/21,720 (0.0%)	not estimable	0 per 1000	
All-cause mortality											
43,448 (1 RCT)	not serious	not serious	not serious	Serious ^b,c^	none	⨁⨁⨁◯ Moderate	4/21,728 (0.0%)	2/21,720 (0.0%)	RR 0.50 (0.09 to 2.73)	0 per 1000	0 fewer per 1000 (from 0 fewer to 0 fewer)
Any serious adverse events											
37,706 (1 RCT)	not serious	not serious	not serious	Serious ^b^	none	⨁⨁⨁◯ Moderate	111/18,846 (0.6%)	126/18,860 (0.7%)	RR 1.13 (0.88 to 1.46)	6 per 1000	1 more per 1000 (from 1 fewer to 3 more)
Related unsolicited serious adverse event											
37,706 (1 RCT)	not serious	not serious	not serious	Serious ^b^	none	⨁⨁⨁◯ Moderate	0/18,846 (0.0%)	4/18,860 (0.0%)	RR 8.99 (0.48 to 167.03)	0 per 1000	0 fewer per 1000 (from 0 fewer to 0 fewer)

CI: Confidence interval; RR: Risk ratio. Explanations: ^a^. The primary and secondary outcomes were reported following the per-protocol analysis approach. ^b^. The 95% confidence intervals for this outcome were wide. ^c^. The 95% confidence interval for this outcome includes the possibility of no effect. ^d.^ The overall certainty of evidence is visually delineated on a four-point scale.

**Table 3 vaccines-10-02065-t003:** GRADE evidence profile denoting the benefits and risks associated with Johnson & Johnson vaccine for the prevention of COVID-19 in adults. The table includes relative and absolute effects along with the overall certainty of the evidence.

Certainty Assessment							Summary of Findings				
Participants (Studies) Follow-Up	Risk of Bias	Inconsistency	Indirectness	Imprecision	Publication Bias	Overall Certainty of Evidence	Study Event Rates (%)		Relative Effect (95% CI)	Anticipated Absolute Effects	
							With Placebo	With Johnson & Johnson		Risk with Placebo	Risk Difference with Johnson & Johnson
Any symptomatic cases											
44,325 (1 RCT)	Serious ^a^	not serious	not serious	not serious	none	⨁⨁⨁◯ ^d^ Moderate	435/22,151 (2.0%)	195/22,174 (0.9%)	RR 0.45 (0.38 to 0.53)	20 per 1000	11 fewer per 1000 (from 12 fewer to 9 fewer)
Symptomatic cases after effective period											
44,325 (1 RCT)	Serious ^a^	not serious	not serious	not serious	none	⨁⨁⨁◯ Moderate	195/22,151 (0.9%)	66/22,174 (0.3%)	RR 0.34 (0.26 to 0.45)	9 per 1000	6 fewer per 1000 (from 7 fewer to 5 fewer)
Severe cases after effective period											
44,325 (1 RCT)	Serious ^a^	not serious	not serious	Serious ^b^	none	⨁⨁◯◯ Low	34/22,151 (0.2%)	5/22,174 (0.0%)	RR 0.15 (0.06 to 0.38)	2 per 1000	1 fewer per 1000 (from 1 fewer to 1 fewer)
COVID-19 deaths											
44,325 (1 RCT)	Serious ^c^	not serious	not serious	Serious ^b^	none	⨁⨁◯◯ Low	5/22,151 (0.0%)	0/22,174 (0.0%)	RR 0.09 (0.01 to 1.64)	0 per 1000	0 fewer per 1000 (from 0 fewer to 0 fewer)
All-cause mortality											
44,325 (1 RCT)	Serious ^c^	not serious	not serious	Serious ^b^	none	⨁⨁◯◯ Low	16/22,151 (0.1%)	3/22,174 (0.0%)	RR 0.19 (0.05 to 0.64)	1 per 1000	1 fewer per 1000 (from 1 fewer to 0 fewer)
Any serious adverse events											
43,783 (1 RCT)	Serious ^c^	not serious	not serious	not serious	none	⨁⨁⨁◯ Moderate	96/21,888 (0.4%)	83/21,895 (0.4%)	RR 0.86 (0.64 to 1.16)	4 per 1000	1 fewer per 1000 (from 2 fewer to 1 more)
Related unsolicited serious adverse event											
43,783 (1 RCT)	Serious ^c^	not serious	not serious	Serious ^b^	none	⨁⨁◯◯ Low	2/21,888 (0.0%)	7/21,895 (0.0%)	RR 3.50 (0.73 to 16.84)	0 per 1000	0 fewer per 1000 (from 0 fewer to 1 more)

CI: Confidence interval; RR: Risk ratio. Explanations: ^a^. Data for this outcome were not available for all participants randomized. ^b^. The 95% confidence intervals for this outcome were wide. ^c^. Data for this outcome were not available for all participants vaccinated. ^d.^ The overall certainty of evidence is visually delineated on a four-point scale.

## Data Availability

The data presented in this study are openly available per our citations below.

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
