# Peer review of "Comparative Benefits and Risks Associated with Currently Authorized COVID-19 Vaccines"

_vaccines, 2022, doi:10.3390/vaccines10122065_

Round 1
Reviewer 1 Report
Albers et al. summarize and compare 5 trials related to the most widely distributed COVID-19 vaccines. The rationale is sound and providing clear comparative data might reduce vaccine hesitance worldwide. The analyzed data covers 5 major trials and some additional FDA documents. The 150k enrolled subjects represent 0.003% of people vaccinated worldwide (5 billion fully vaccinated people) but are of course well-characterized. Why are there no additional (follow-up) trials included (or is this data not available)? Please motivate.
The results are systematically reported. However, in this way it remains highly unappealing to read. I would prefer to add a summary table and a visual representation (i.e. a graph) to indicate all these results per topic (e.g. COVID-19 related mortality).
Please avoid using Covid-19 (consistently use COVID-19).
Sputnik assessment: twice the vaccine arm is mentioned, please amend.
Alinea lines 374: please change ‘do’ to ‘do not’ to the first sentence.
Discussion in alinea 386: please add a comparison for MIS-C post vaccination and post-COVID-19 to engage also the adolescent/pediatric population.
Author Response
We want to thank you for your productive peer review.
Comment 1: Why are there no additional (follow-up) trials included (or is this data not available)? Please motivate.
Response: Thank you for your comments. Please note that the study was conducted using the systematic review method that requires a cut-off date for inclusion of all studies. At the time of the cut-off only 5 vaccines had publicly available data either in the form of peer-reviewed publication or as a FDA approval document. That is why, we did not include trials that were completed and reported later. Thank you.
Comment 2: The results are systematically reported. However, in this way it remains highly unappealing to read. I would prefer to add a summary table and a visual representation (i.e. a graph) to indicate all these results per topic (e.g. COVID-19 related mortality
Response: Thank you for your comments. This comment appears to be an example of miscommunication. Please note, that the tables summarize all reported benefit and risks associated with included trials in the form of GRADE evidence profiles which is currently the standard for reporting of benefits and risks for any guideline as well. Furthermore, for all critical outcomes we also have provided forest plots as figure. Given the limited number of figures and tables allowed as part of the manuscript, we decided to illustrate figures as supplementary material instead of using it in the main document. Thank you.
Comments 3 – 5:
Please avoid using Covid-19 (consistently use COVID-19).
Sputnik assessment: twice the vaccine arm is mentioned, please amend.
Alinea lines 374: please change ‘do’ to ‘do not’ to the first sentence.
Response: Thank you for your assistance with these typographical errors. Points 3,4,5 have been addressed per your direction.
Comment 6: Discussion in alinea 386: please add a comparison for MIS-C post vaccination and post-COVID-19 to engage also the adolescent/pediatric population.
Response: We thank you for your suggestion and see the potential merit of this inclusion. However, at the time of this study the results from pediatric populations were not available. For this reason, we feel the inclusion of the pediatric populations is beyond the scope of work for this study.
Reviewer 2 Report
The manuscript by Albers et al reviewed the efficacy and safety data of 5 approved vaccines and concluded that the 5 approved vaccines are effective and safe. Although this review provided a quick glance into the efficacy and safety data of these vaccines, however, no impressive findings were found and the importance of this review is also shadowed by the many homogeneous reviews. Currently I do not feel it is necessary to publish this article in Vaccines. Besides, I see some mistakes that harmed the overall quality of this manuscript.
1: Article data is incomplete and tables are cluttered and disordered. Missing Figure1a, Figure 1b, Figure 1c, supplemental table 1, and supplemental table 2, etc.
2: There are also some other minor mistakes, such as,
1) Line 2, “Risk” should be “Risks”.
2) Line 76, Decryption of ITT.
3) Line 79, “vaccine” should be “placebo”.
4) Line 123, the same.
5) Line 97, “Figure 1a” should be “Figure 2a”.
6) Line 103, “Figure 1b” should be “Figure 2b”.
7) Lack of figure legends.
Author Response
General comment: The manuscript by Albers et al reviewed the efficacy and safety data of 5 approved vaccines and concluded that the 5 approved vaccines are effective and safe. Although this review provided a quick glance into the efficacy and safety data of these vaccines, however, no impressive findings were found and the importance of this review is also shadowed by the many homogeneous reviews. Currently I do not feel it is necessary to publish this article in Vaccines. Besides, I see some mistakes that harmed the overall quality of this manuscript.
Response: Thank you for your comments. We respectfully disagree. The goal of this manuscript along with GRADE evidence profile table is to provide the benefits and risks associated with COVID-19 vaccines in one place, transparently, to facilitate shared decision making by informing concisely patients, physicians and policy makers. Having an accurate representation of the evidence is critical to this effort. Thank you.
Comment 1: Article data is incomplete, and tables are cluttered and disordered. Missing Figure1a, Figure 1b, Figure 1c, supplemental table 1, and supplemental table 2, etc.
Response: We thank you for pointing out these typographical errors made when transferring our submission materials to the Vaccines pre-formatted template. We are open to adjusting the format of manuscript, and this is our current interpretation of the journal requirements and encouraged use of the journals recommended template. We have adjust the language discussing all figures in the text to accurately represent the fact they are indeed supplemental figures. Supplemental figures were added.
Comment 2: There are also some other minor mistakes, such as:
- Line 2, “Risk” should be “Risks”.
- Line 76, Decryptionof ITT.
- Line 79, “vaccine” should be “placebo”.
- Line 123, the same.
- Line 97, “Figure 1a” should be “Figure 2a”.
- Line 103, “Figure 1b” should be “Figure 2b”.
- Lack of figure legends.
Response: We thank you for finding and assisting with these typographical errors. Each has been specifically addressed per your direction in text.
Reviewer 3 Report
The review article is well written and the authors did an excellent job in collating data from a variety of sources, and analyzing it for the general scientific community. I do not have any comments or corrections but congratulate the authors for a job well-done.
Author Response
We would like to thank you for your support and emphatic interest in our work.
Round 2
Reviewer 1 Report
The authors have adopted the suggestions (in part).
I would recommend the authors to indicate the limitations of their research (limited evaluated trial period - this is a highly rapidly evolving field - , no pediatric data etc).
Author Response
Reviewer 1 has suggested a more direct statement on the limitations of our study.
We thank you kindly for your suggestion. We have added a paragraph dedicated to articulating the limitations of this analysis in the discussion.
Reviewer 2 Report
The manuscript by Albers et al reviewed the efficacy and safety data of 5 approved vaccines and concluded that the 5 approved vaccines are effective and safe. Although this review provided a GRADE evidence profile table to analyse the efficacy and safety of these vaccines, however, no impressive findings were found.
Author Response
Reviewer 2 summarized our research.
We thank you kindly for taking an interest in our work, and your time in the review process.
Round 3
Reviewer 1 Report
I have no further comments.